# Mapping Quantitative Trait Loci for 1000-Grain Weight in a Double Haploid Population of Common Wheat

**DOI:** 10.3390/ijms21113960

**Published:** 2020-05-31

**Authors:** Tao Liu, Lijun Wu, Xiaolong Gan, Wenjie Chen, Baolong Liu, George Fedak, Wenguang Cao, Dawn Chi, Dengcai Liu, Huaigang Zhang, Bo Zhang

**Affiliations:** 1Northwest Institute of Plateau Biology, Key Laboratory of Adaptation and Evolution of Plateau Biota (AEPB), Chinese Academy of Sciences, Xining 810008, China; liutao415@mails.ucas.ac.cn (T.L.); wljd126@126.com (L.W.); ganxiaolong16@mails.ucas.ac.cn (X.G.); cwj60905@163.com (W.C.); blliu@nwipb.cas.cn (B.L.); 2University of Chinese Academy of Sciences, Beijing 100049, China; 3Qinghai Province Key Laboratory of Crop Mol. Breeding, Xining 810008, China; 4Ottawa Research and Development Centre, Agriculture and Agri-Food Canada, 960 Carling Avenue, Ottawa, ON K1A 0C6, Canada; George.Fedak@Canada.ca (G.F.); wenguang.Cao@Canada.ca (W.C.); Dawn.Chi@Canada.ca (D.C.); 5Triticeae Research Institute, Sichuan Agricultural University, Chengdu 611130, China; dcliu7@yahoo.com

**Keywords:** DH population, wheat, QTL analysis, TGW, wheat55k SNP array

## Abstract

Thousand-grain weight (TGW) is a very important yield trait of crops. In the present study, we performed quantitative trait locus (QTL) analysis of TGW in a doubled haploid population obtained from a cross between the bread wheat cultivar “Superb” and the breeding line “M321” using the wheat 55-k single-nucleotide polymorphism (SNP) genotyping assay. A genetic map containing 15,001 SNP markers spanning 2209.64 cM was constructed, and 9 QTLs were mapped to chromosomes 1A, 2D, 4B, 4D, 5A, 5D, 6A, and 6D based on analyses conducted in six experimental environments during 2015–2017. The effects of the QTLs *qTgw.nwipb-4DS* and *qTgw.nwipb-6AL* were shown to be strong and stable in different environments, explaining 15.31–32.43% and 21.34–29.46% of the observed phenotypic variance, and they were mapped within genetic distances of 2.609 cM and 5.256 cM, respectively. These novel QTLs may be used in marker-assisted selection in wheat high-yield breeding.

## 1. Introduction

Bread wheat is one of the world’s major grain crops. Due to an increasing population [1], the development of new high-yielding varieties remains the primary goal of wheat-breeding programs [2]. The final grain yield is a complex trait that is often strongly affected by genetic and environmental factors. In cereal crops, the thousand-grain weight (TGW) is an important yield component, and an increased TGW is key for further increasing the grain yield. In addition, TGW is more stably inherited than the overall final production [3,4,5]. 

A genetic map of molecular markers is extremely useful for plant breeding; with an increasing density of molecular markers in genetic maps, a quantitative trait locus (QTL) analysis has been widely used to analyze specific yield-related traits in bread wheat. In recent decades, polymorphic markers such as microsatellite markers [6,7], RAPD (random amplified polymorphic DNA) [8,9], and AFLP (amplified fragment length polymorphism) [9] have been used to construct genetic maps. Many major QTLs related to TGW have been mapped to almost all wheat chromosomes, with the exception of 6D. Campbell et al. (1999) identified three QTLs for TGW on chromosomes 1A, 1B, and 7A [10]. Varshney et al. (2000) found multiple QTLs on chromosomes 1A, 1D, 2B, 4B, 5B, 6B, 7A, and 7D controlling TGW [6]. Huang et al. (2003) discovered eight QTLs on chromosomes 2A, 2D, 4D, 5A, 7B, and 7D regulating TGW in a BC_2_F_2_ population [11]. Two prominent TGW QTLs were detected on chromosomes 3D and 4A in a study by McCartney et al. (2005) [8]. Ramya et al. (2010) detected six QTLs on chromosomes 1A, 2B, 2D, 5A, 5B, and 5D associated with the TGW trait [12]. Other QTLs for TGW have been identified on other chromosomes [5,13,14,15,16,17,18,19]. Due to the large size of the wheat genome, the development and application of these markers in wheat are time-consuming and expensive, and the genetic distance between the markers in the maps is relatively large, so map-based gene cloning is difficult to perform [20]. To date, *TaGW2-6A* and *TaFlo2-A1* are the only two genes controlling the grain weight that have been cloned [21,22].

A high-density genetic map is more effective and essential for QTL analysis in cereal crops [23,24]. However, single-nucleotide polymorphisms (SNPs) are the most promising molecular markers [25], showing a high abundance and a uniform distribution throughout the genome, and have been increasingly applied in plants in recent years [26]. Low mutation rates and stable genetics are key characteristics of SNPs. The QTLs obtained from SNPs present a high resolution and can be used for marker-assisted selections to improve selection accuracy and breeding efficiency [27]. With the development of sequencing technology, various wheat SNP array chips have been developed and utilized for QTL analysis, such as the Wheat 9 K, Wheat 90 K, Wheat 660 K, and Wheat 820 K chips [23,28,29,30]. The Wheat 55 K chip developed by the Chinese Academy of Agricultural Sciences contains 53,063 SNP markers, which were carefully selected from the Wheat 660 K array. Liu et al. (2018) constructed a high-density genetic map and identified a novel major QTL for a productive tiller number by using Wheat 55 K SNP array chips [20]. Ren et al. (2018) utilized the Wheat 55 K SNP array to perform a QTL analysis for the tiller number in a wheat recombinant inbred line population and identified *cqTN-2D.2* as a major QTL [31]. Compared with other SNP arrays, the Wheat 55 k SNP array is more efficient and lower in cost. 

In the present study, nine TGW QTLs, including two major QTLs (*qTgw.nwipb-4DS* and *qTgw.nwipb-6AL*) were detected in a doubled haploid (DH) population using a wheat 55 k SNP array and three years of TGW data from three environments on the Qinghai Plateau. The newly detected *qTgw.nwipb-4DS* and *qTgw.nwipb-6AL* loci are expected to be valuable for map-based cloning and wheat improvement.

## 2. Results

### 2.1. Genetic Map Construction

According to the minor allele frequency (frequency < 0.3) and marker polymorphism between the two parents, approximately 18-k SNPs were selected from the Wheat 55 k chip. After filtering the SNPs with missing rates > 10% and distortion values < 0.01, a high-density genetic map containing 15,001 SNP markers spanning 2209.64 cM was constructed (Table 1), with an average of 0.147 cM per SNP locus. Among these SNPs, 6876 (45.84%), 6374 (42.49%), and 1751 (11.67%) were mapped to genomes A, B, and D, respectively (Appendix A).

The 15,001 markers were divided into 773 bins, with an average distance of 2.859 cM between two bins, and these bins were mapped to 25 linkage groups for the 21 chromosomes of wheat (Table 1). Two linkage groups were constructed for chromosomes 3B, 4D, 5D, and 6D. Among these bins, 263 bins included only one marker, and the largest bins contained 1431 SNPs (Appendix A and Appendix A). Most of the bins were distributed on genome A (37.39%) and genome B (36.22%), while only 26.39% of bins were mapped to genome D. The length of the linkage group maps ranged from 13.11 cM (3B-2) to 158.98 cM (3A). The bin numbers in the maps ranged from 10 to 55. The average genetic distance between contiguous bin markers ranged from 1.31 cM to 4.72 cM.

The flanking sequences of the SNPs were used for BLAST searches with the IWGSC (International Wheat Genome Sequencing Consortium) refseqv1.0. The order of most of the SNP markers distributed in the present genetic map were consistent with the published wheat genome (Figure 1 and Appendix A). Among 15,001 SNP markers, 1304 SNP markers (8.69%) showed the best hits to the CDS (coding sequence) of Chinese spring, and these SNPs were considered to be coding-region SNPs (Appendix A).

### 2.2. Phenotypic Statistical Analysis

The 1000-grain weight values of Superb and M321 from the six environments were 47.40 (g) ± 1.44 (g) and 38.76 (g) ± 1.30 (g), respectively, indicating a significant difference in the TGW trait between Superb and M321 (Appendix A and Appendix A). The TGW trait showed an obviously normal distribution in the DH population (Appendix A). The mean TGW of the DH populations in six environments was used for correlation analysis, and the results showed highly significant positive correlations (Appendix A). The heritability value for TGW in these populations was 0.59 (Table 2).

### 2.3. QTL Analysis for TGW 

The biparental populations (BIP) function of IciMapping with the inclusive composite interval mapping (ICIM) module was applied to analyze the TGW trait of the DH population. After a 1000-times permutation test, the LOD (likelihood of odds) threshold was determined to be 3.22. Nine QTLs were detected in six environments, and they were mapped on chromosomes 1A, 2D, 4B, 4D, 5A, 5D, 6A, and 6D (Figure 2 and Table 3). The LOD values of each QTL ranged from 3.29 to 12.19, and the additive effect ranged from −2.139 to 2.605. The phenotypic variation explained by these QTLs ranged from 6.81% to 32.43%. Considering the physical locations of the SNPs, these QTLs were named *qTgw.nwipb-1AS, qTgw.nwipb-2DS*, *qTgw.nwipb-2DL*, *qTgw.nwipb-4BS*, *qTgw.nwipb-4DS*, *qTgw.nwipb-5AL*, *qTgw.nwipb-5DS*, *qTgw.nwipb-6AL,* and *qTgw.nwipb-6DS*. Among these QTLs, *qTgw.nwipb-4DS* and *qTgw.nwipb-6AL* could be detected in multiple environments in different years.

Interestingly, *qTgw.nwipb-4DS* and *qTgw.nwipb-6AL* explained 15.31%–32.43% and 21.34%–29.46% of the observed phenotypic variation, respectively, and both of them exhibited a relatively high LOD value in every environment, indicating that they were stable and robust QTLs. *qTgw.nwipb-4DS* was located in the interval of AX-95683531--AX-108924542, within a 2.609-cM genetic distance from 7.652 cM to 10.261 cM, and *qTgw.nwipb-6AL* was located in AX-108982634--AX-110577250, within a 5.256-cM region from 62.382 cM to 67.638 cM. The additive effect was positive in *qTgw.nwipb-4DS*, indicating that the alleles from M321 increased the TGW, while negative in *qTgw.nwipb-6AL*, demonstrating that the Superb alleles enhanced the TGW. 

A combined QTL-by-environment interaction analysis performed with the multienvironmental trials (MET) module and the LOD threshold value of 6.23, seven QTLs were identified and mapped then to chromosomes 1A, 2D, 4B, 4D, 5A, 6A, and 6D (Figure 2 and Table 4). The two stable QTLs described above were also found to be robust in this module. The LOD values for *qTgw.nwipb-4DS* and *qTgw.nwipb-6AL* were 33.73 and 36.99, respectively, and the LOD (A) and LOD (AE) values were 26.70 and 7.03, respectively, for *qTgw.nwipb-4DS* and 34.29 and 2.70 for *qTgw.nwipb-6AL*. Additionally, the PVE and PVE (AE) explained by *qTgw.nwipb-4DS* were 22.24% and 4.57%, respectively, while those explained by *qTgw.nwipb-6AL* were 26.96% and 3.41%. The flanking markers of the QTLs were physically mapped on the chromosomes, these QTLs span 5.28 Mb (1AS), 102.18 Mb (2DS), 3.85 Mb (4BS), 65.81 Mb (4DS), 1.67 Mb (5AL), 5.97 Mb (6AL), and 13.64 Mb (6DS), respectively. Of them, these two stable QTLs span 65.81 Mb (*qTgw.nwipb-4DS*) and 5.97 Mb (*qTgw.nwipb-6AL*, located on the near-distal of chromosomes 4DS and 6AL, respectively (Appendix A).

## 3. Discussion

Grain yield is the most important trait in wheat-breeding programs, which was a continuous variation trait controlled by multiple quantitative trait loci [33]. Most of grain yield-related traits, such as the flowering period, and biological, as well as abiotic, resistance, are controlled by many genes with low heritability. TGW is an important component of grain yield, which is a high-heritability yield trait, and the influence of the environment on it is significantly lower or even insensitive than other yield-related traits [34]. TGW is often positively correlated with the crop yield [3], indicating that it is possible to indirectly improve 1000-grain weight to increase the grain yield. To meet the increasing demand for wheat production, an increasing number of valuable TGW genes (QTLs) should be identified to increase the wheat yield.

In this study, seven QTLs of the TGW trait were identified in six environments, six of which were located in genomes A and D and explained more than 70% of the observed phenotypic variation. *qTgw.nwipb-4DS* and *qTgw.nwipb-6AL* were detected in more than four environments. *qTgw.nwipb-6AL* was expressed in five environments, with the exception of XN2017. Continuous rainfall from June to September 2017 caused some of the DH lines to undergo preharvest sprouting in Xining, causing the 1000-grain weight data distribution to deviate from those in the other experimental environments, which might be the main reason why *qTgw.nwipb-6AL* was not detected in XN2017.

PVE is the sum of the PVE (A) and PVE (AE). The PVE (A) and PVE (AE) values of these QTLs ranged from 2.14 to 23.55 and 0.20 to 4.57, respectively (Table 4). The values of the PVE (AE) were much lower than those of the PVE (A), indicating that the QTLs were less affected by environmental factors and implying that *qTgw.nwipb-4DS* and *qTgw.nwipb-6AL* were stable QTLs.

In this research, *qTgw.nwipb-4DS* and *qTgw.nwipb-6AL* were identified, and both were found to be major effective and robust QTLs; approximately 50% of the observed phenotypic variation was explained by these two QTLs together. Several QTLs in wheat have been reported to be effective for TGW improvement [34], although they have rarely been reported located on chromosome 6A compared with the other chromosomes [16,18,35]. By using a recombinant inbred line (RIL) population, Su et al. (2011) fine-mapped the gene *TaGW2-6A* (genomic location from 237,735,006 bp to 237,759,304 bp) near the centromere of 6A [21]. In our work, *qTgw.nwipb-6AL* was found to be physically mapped between 573,480,239 bp and 579,446,410 bp on the end of 6AL, and we conclude that it is likely to be a novel QTL with a high LOD and PVE.

In rice, the RING-type E3 ubiquitin ligase OsGW2 has been shown to negatively affect the grain width. The wheat homologue *TaGW2* has been mapped to chromosome 6A. The interval of *qTgw.nwipb-6AL* contains 71 predicted genes based on IWGSC RefSeq v1.0 (Appendix A). Among the 71 genes, *TraesCS6A01G343600* encodes the E3 ubiquitin-protein ligase (from 577,780,417 bp to 577,781,984 bp). Therefore, it may be considered as a potential candidate gene for *qTgw.nwipb-6AL*.

The flanking markers of the QTLs detected on the genetic maps in the current study were mostly consistent with their corresponding positions on the chromosomes. The additive effect was positive in *qTgw.nwipb-4DS*, indicating that the alleles from M321 increased the TGW. However, the flanking physical positions for the flanking markers of *qTgw.nwipb-4DS* were 28,448,472 bp and 94,254,122 bp on the short arm of 4D. It was spanned 65.81 Mb, with only 2.609-cM genetic distance. We preliminarily inferred that chromosome fragment translocation might be occurring during distant hybridization. M321 is a wheat line derived from Superb/*Triticum monococcum*//Fukuho. *T. monococcum* was the donor of the A^m^ genome. Fukuho is a Japanese wheat cultivar with an AABBDD genome. Parental hybrids producing F_1_ plants with different ploidies will produce unequal numbers of univalent bodies during meiosis, leading to chromosome segregation disorders [36]. Translocation might have occurred on chromosomes 4D during the development process of M321, resulting in the variation that was retained in the DH population. Future work should be carried out to confirm the chromosomal structural variation of 4D in M321 and identify the candidate genes of the two major effective QTLs. 

## 4. Materials and Methods 

### 4.1. Plant Materials

A doubled haploid (DH) population including 85 lines derived from a cross between “Superb” and “M321” and was produced by the maize pollination method. “Superb” is a Canadian Western red spring cultivar with a relatively high harvest yield and thousand-grain weight (TGW) and multiple tillers [37,38,39]. Fukuho-Komugi is a Japanese spring cultivar. *T. monococcum* (A^m^A^m^) 10-1 is the A^m^ genome donor. M321 is a spring wheat line derived from a Superb/*T. monococcum* 10-1/Fukuho-Komugi cross with a low thousand-grain weight (TGW); it was developed by Dr. George Fedak in the 1990s.

The DH population and its parents were planted in Delingha (DLH) in the Haixi Mongol and Tibetan Autonomous Prefecture, Qinghai Province (97.37° E, 37.37° N) (2015 and 2017), Liming Village in Haidong (HD) City, Qinghai Province (102.09° E, 36.47° N) (2016 and 2017), and Changning Village in Xining (XN) City, Qinghai Province (101.74° E, 36.56° N) (2016 and 2017). Each DH line and the parents were in a single 2-meter row with 0.2 m between rows, and the sowing density was 100 seeds per row. Nitrogen and superphosphate fertilizers were applied at a rate of 80 and 100 kg/ha, respectively, at sowing. Field management was performed according to the common practices. Thousand-grain weight (TGW) was evaluated by weighing 1000 kernels with precision of 0.01 g in the laboratory after harvest. TGW data were statistically analyzed using SPSS 18.0.

### 4.2. Molecular Genotyping and Genetic Linkage Map Construction

The two parents and each DH line of the population were sprouted in dishes for 2 weeks, and genomic DNA was extracted from young leaves using the Plant Genomic DNA Kit (TIANGEN Biotech, Beijing, China). A total of 1 μL of DNA solution was used to check the DNA quality in 10-g/L agarose gels by electrophoresis, and the DNA concentration was measured with a NanoDrop 2000 C spectrophotometer (Thermo Scientific, Wilmington, DE, USA). The DNA of the DH population was genotyped with the 55-k iSelect single-nucleotide polymorphism (SNP) genotyping assay, which contains 53,063 markers (CaptialBio Technology, Beijing, China).

The SNP data were considered to be missing data if they were heterozygous. The markers that were identical in the two parents were rejected and then converted to SNP data in the format required by IciMapping 4.1 [40]. The BIN function of IciMapping 4.1 was used to filter the SNP markers by removing the data exhibiting more than 10% missing data and a distortion value of less than 0.01. Each bin included no less than one marker. A Bin ID value -1 indicated that a marker had been deleted, and a positive value indicated that markers had been retained. Then, markers were selected according to the lowest missing rate from each bin. By using JoinMap 4.0, the markers were ordered into corresponding groups with LOD scores ranging from 2 to 10, and each group was then used to construct a linkage map by using the Kosambi mapping function with an LOD ≥ 5 [20,41]. To obtain the physical locations of the SNPs and divide the long and short chromosome arms of the genetic map, the flanking sequences were subjected to BLAST searches with the IWGSC wheat contig sequences (http://www.wheatgenome.org/). All of the genetic maps generated in the present study were drawn by using Mapchart 2.0 [42].

### 4.3. Statistical Analysis and QTL Mapping

The TGW phenotypic data from each environment were statistically analyzed using SPSS 18.0. The formula used to compute heritability was derived from Hu et al. (2017) [43]. IciMapping 4.1 was used for QTL analysis. The QTLs for TGW from every environment were detected in the biparental populations (BIP), in which an inclusive composite interval mapping (ICIM) module with an LOD score value over 3.0 was used. QTLs and environmental interactions were studied in multienvironment trials (METs). The mapping parameters were set as follows: QTL walking speed of 1.00 cM, stepwise regression probability threshold value of 0.001, and the threshold LOD was determined by permutation tests (Times = 1000 and Type I Error = 0.05). The international customary rule was applied to nominate QTLs. “Tgw”, “nwipb”, “L”, and “S” represent the “thousand-grain weight”, “Northwest Institute of Plateau Biology, CAS”, the “long arm of the chromosome”, and the “short arm of the chromosome”, respectively.

## 5. Conclusions 

In summary, nine TGW QTLs were identified in this study, including the major and robust novel loci *qTgw.nwipb-4DS* and *qTgw.nwipb-6AL*, with genetic distances of 2.609 cM and 5.256 cM, respectively. *TraesCS6A01G343600* encodes the E3 ubiquitin-protein ligase; it may be considered as a potential candidate gene for *qTgw.nwipb-6AL*. These two novel QTLs may be used in marker-assisted selections in wheat high-yield breeding. Our findings provided new data on QTLs related to thousand-grain weight in wheat and will enhance the understanding of the genetic basis of TGW traits.

## Figures and Tables

**Figure 1 ijms-21-03960-f001:**
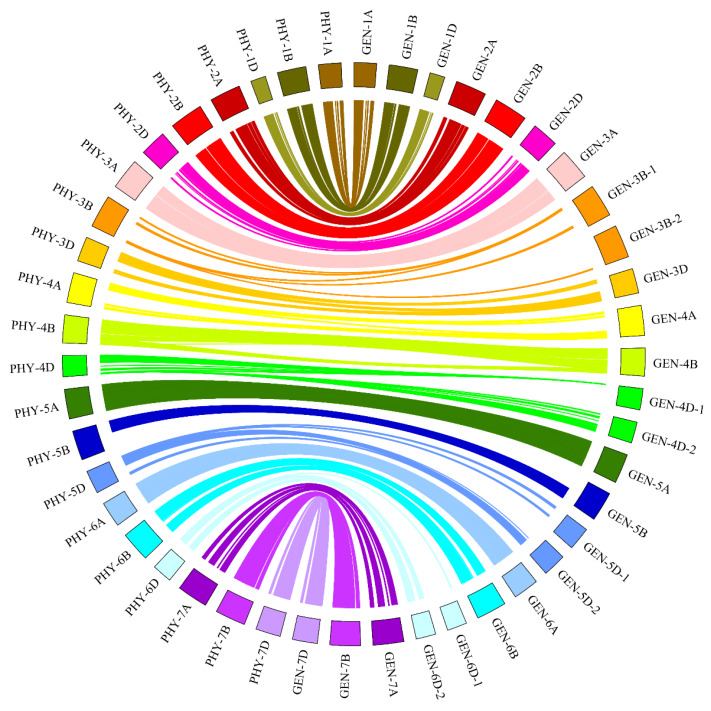
The diagrammatic of the relationship between single-nucleotide polymorphism (SNP) markers in wheat genetic and physical maps. GEN-1A to GEN-7D denote the 25 linkage groups that belong to the 21 chromosomal genetic maps of wheat, and PHY-1A to PHY-7D show the 21 chromosomal physical maps of wheat.

**Figure 2 ijms-21-03960-f002:**
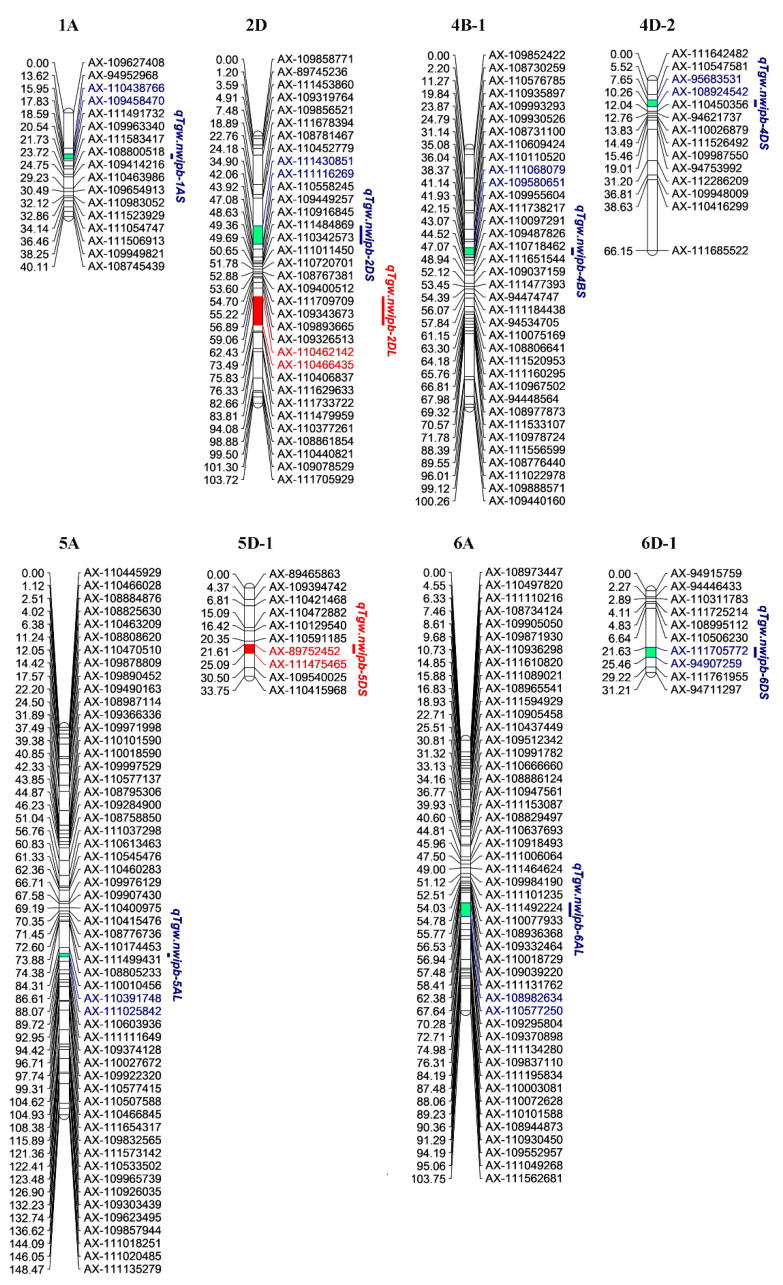
Quantitative trait locus (QTL) for 1000-grain weight trait of the doubled haploid (DH) population. Genetic distances are annotated on the left of each chromosome, and the markers names are annotated on the right. The QTLs with red color show that they were identified by the biparental populations (BIP) function, and those with purple color denote that they were detected by both the BIP function and multienvironmental trials (MET) function.

**Table 1 ijms-21-03960-t001:** Distribution of the mapped markers in the genetic map. SNP: single-nucleotide polymorphisms.

Chr.	Group	Length (cM)	SNP Markers	cM per	Number of	cM per
				SNP Marker	Bin Markers	Bin Marker
1A	1	40.11	535	0.07	17	2.36
1B	1	95.80	726	0.13	31	3.09
1D	1	60.99	527	0.12	16	3.81
2A	1	86.71	1613	0.05	32	2.71
2B	1	146.21	1116	0.13	52	2.81
2D	1	103.72	194	0.53	34	3.05
3A	1	158.98	897	0.18	53	3.00
3B	1	67.69	146	0.46	33	2.05
	2	13.11	31	0.42	10	1.31
3D	1	112.02	93	1.20	27	4.15
4A	1	110.26	650	0.17	41	2.69
4B	1	100.26	1496	0.07	36	2.79
4D	1	21.17	56	0.38	10	2.12
	2	66.15	48	1.38	14	4.72
5A	1	148.47	896	0.17	55	2.70
5B	1	92.68	549	0.17	44	2.11
5D	1	33.75	26	1.30	10	3.38
	2	90.25	75	1.20	33	2.73
6A	1	103.75	1678	0.06	48	2.16
6B	1	119.78	955	0.13	42	2.85
6D	1	31.21	14	2.23	10	3.12
	2	67.26	89	0.76	18	3.74
7A	1	143.21	607	0.24	43	3.33
7B	1	112.93	1355	0.08	32	3.53
7D	1	83.17	629	0.13	32	2.60
Total	25	2209.64	15,001	0.15	773	2.86

**Table 2 ijms-21-03960-t002:** Value of thousand-grain weight (TGW) (g) trait in parents and the doubled haploid (DH) lines across all environments.

Environment	Parents		DH Lines								
	S	M	Range	Min.	Max.	Mean	SD	CV (%)	Sk.	Ku.	h^2^
HX2015	48.23	39.18	19.72	32.20	51.95	40.59	4.46	19.93	0.056	−0.631	0.59
HD2016	48.39	39.90	21.54	30.90	52.44	43.71	4.33	18.78	−0.178	−0.175	
XN2016	46.64	38.46	20.37	30.11	50.48	41.54	4.18	17.45	−0.461	−0.026	
HD2017	48.30	40.07	20.88	30.00	50.88	42.04	4.17	17.35	−0.150	−0.112	
XN2017	44.78	36.49	20.02	31.08	51.10	41.09	5.04	25.44	−0.070	−0.785	
HX2017	48.08	38.46	18.78	31.70	50.48	41.85	4.28	18.28	−0.345	−0.466	

S, “Superb”; M, “M321”; SD, standard deviation; CV, coefficient of variation; Sk., skewness; Ku., kurtosis; and h^2^, heritability.

**Table 3 ijms-21-03960-t003:** Quantitative trait locus (QTL) for the TGW traits in different environments.

QTL	Pos.(cM)	Environment	Chr.	Interval (cM)	Flanking markers	LOD	Add	PVE (%)	Physical pos. (Mb)	Near Locus in Previous Studies
*qTgw.nwipb-1AS*	16.890	HD2017	1A	15.954–17.833	AX-110438766--AX-109458470	3.68	−1.204	8.09	20.02–25.31	QGw.ccsu-1A.1 [5]
*qTgw.nwipb-2DS*	40.270	DLH2015	2D	34.900–42.060	AX-111430851--AX-111116269	4.98	−1.698	14.54	243.79–345.97	
*qTgw.nwipb-2DL*	70.414	XN2017	2D	62.425–73.487	AX-110462142--AX-110466435	4.44	−1.449	9.81	585.69–601.05	
*qTgw.nwipb-4BS*	40.678	DLH2017	4B	38.368–41.140	AX-111068079--AX-109580651	4.24	1.692	12.75	25.35–29.20	QTKW.caas-4BS [32]
*qTgw.nwipb-4DS*	8.956	DLH2015	4D	7.652–10.261	AX-95683531--AX-108924542	5.16	1.744	15.31	28.44–94.25	
		XN2016	4D	7.652–10.261	AX-95683531--AX-108924542	5.12	1.789	17.17		
		HD2017	4D	7.652–10.261	AX-95683531--AX-108924542	9.44	2.103	24.65		
		XN2017	4D	7.652–10.261	AX-95683531--AX-108924542	12.19	2.605	32.43		
*qTgw.nwipb-5AL*	87.339	XN2017	5A	86.608–88.069	AX-110391748--AX-111025842	3.38	1.235	6.97	594.37–596.03	QTKW.caas-5AL [32]
*qTgw.nwipb-5DS*	22.651	HD2016	5D	21.606–25.089	AX-89752452--AX-111475465	3.59	1.415	13.16	41.82–44.93	
*qTgw.nwipb-6AL*	64.353	DLH2015	6A	62.382–67.638	AX-108982634--AX-110577250	7.18	−2.113	22.43	573.48–579.45	
		XN2016	6A	62.382–67.638	AX-108982634--AX-110577250	6.91	−2.139	24.44		
		HD2016	6A	62.382–67.638	AX-108982634--AX-110577250	7.38	−2.119	29.46		
		DLH2017	6A	62.382–67.638	AX-108982634--AX-110577250	7.61	−2.404	25.97		
		HD2017	6A	62.382–67.638	AX-108982634--AX-110577250	8.41	−1.956	21.34		
*qTgw.nwipb-6DS*	23.301	XN2017	6D	21.626–25.455	AX-111705772--AX-94907259	3.29	−1.195	6.81	10.91–24.55	

Pos., position; Chr., chromosome; Add, additive effect—positive values indicate that M321 alleles increased the TGW trait, and negative values indicate that the Superb alleles increased it; LOD, likelihood of odds; and PVE (%), proportion of phenotypic variation of the corresponding QTL.

**Table 4 ijms-21-03960-t004:** QTL analysis for TGW from multienvironmental trials (MET).

QTL	Chr.	Interval (cM)	LOD	LOD (A)	LOD (AE)	PVE	PVE (A)	PVE (AE)	Add
*qTgw.nwipb-1AS*	1A	15.954–17.833	9.06	7.97	1.09	5.02	4.62	0.40	−0.792
*qTgw.nwipb-2DS*	2D	34.900–42.060	9.11	7.01	2.11	6.06	4.15	1.90	−0.753
*qTgw.nwipb-4BS*	4B	38.368–41.140	6.69	3.66	3.04	4.82	2.14	2.68	0.545
*qTgw.nwipb-4DS*	4D	7.652–10.261	33.73	26.70	7.03	22.24	17.66	4.57	1.551
*qTgw.nwipb-5AL*	5A	86.608–88.069	10.92	10.20	0.71	6.06	5.86	0.20	0.914
*qTgw.nwipb-6AL*	6A	62.382–67.638	36.99	34.29	2.70	26.96	23.55	3.41	−1.795
*qTgw.nwipb-6DS*	6D	21.626–25.455	6.67	5.67	1.00	3.81	3.28	0.53	−0.669

LOD (A) and LOD (AE): indicate the LOD value for additive and dominance effects and LOD score for additive and dominance by environment effects, respectively. PVE (A) and PVE (AE) represent the phenotypic variation explained by additive and dominance effects and additive and dominance by environment effects, separately.

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
