# Peer review of "Mapping Quantitative Trait Loci for 1000-Grain Weight in a Double Haploid Population of Common Wheat"

_ijms, 2020, doi:10.3390/ijms21113960_

Round 1

Reviewer 1 Report

This is a valuable research with an important grain crop. You should consider improving the title of your manuscript to make it more appropriate. Your research is progressing with Thousand Grain Weight not really with grain yield directly.

Line 38-39 please rewrite, not making proper sense, be specific and direct.

Please check the comments on line 107, the findings not clearly presented in Table 2, more over, clear statistical analysis like paired t-test is necessary for making such conclusion. 

In Table 3, position of QTL refers what? Please clearly mention base pair or what? 

Line 198, it would be a good idea to add details how and when DH population was developed. 

Line 208, add experimental designs and description of trials.

In conclusion, add up few major findings and discuss future research direction.

Author Response

Point 1: This is a valuable research with an important grain crop. You should consider improving the title of your manuscript to make it more appropriate. Your research is progressing with Thousand Grain Weight not really with grain yield directly.

Response 1: Thanks for your reasonable suggestion. The title “Mapping quantitative trait loci for grain yield in a double haploid population of common wheat” was changed as “Mapping quantitative trait loci for 1000-grain weight in a double haploid population of common wheat” in our revised manuscript.

Point 2: Line 38-39 please rewrite, not making proper sense, be specific and direct.

Response 2: Thanks for your advice. We deleted this sentence from our manuscript.

Point 3: Please check the comments on line 107, the findings not clearly presented in Table 2, moreover, clear statistical analysis like paired t-test is necessary for making such conclusion. 

Response 3: As the reviewer mentioned statistical analysis like paired t-test is very necessary, we did t-test in Table S5, because there are not enough space to include t-test in Table 2.  

Point 4: In Table 3, position of QTL refers what? Please clearly mention base pair or what? 

Response 4: Thank you for suggestion, we added the “cM” as unit in Table 3.

Point 5: Line 198, it would be a good idea to add details how and when DH population was developed. 

Response 5: We added the details of the DH population production in line 219-220 of clean version.

Point 6: Line 208, add experimental designs and description of trials.

Response 6: Thank you for the suggestion. We added the field experimental designs in line 224-229 in our clean version.

Point 7: In conclusion, add up few major findings and discuss future research direction.

Response 7: We added up few major findings and future research direction in line 265-269.

Reviewer 2 Report

The manuscript provides new data on QTLs related to thousand grain weight (TGW) in wheat. Generally approach of QTL mapping on biparental DH population, phenotyping in multiple locations and chip based genotyping are good and promising. However, the main pitfalls are:

  1. low number of plants (~85) reduces precision of genetic mapping and in effect discrepancies between genetic and physical maps are so significant that no candidate genes can be proposed; to minimize these effects construction of genetic map should be revised and generally most of markers not colinear with physical map should be possibly removed (7A2, 1B2, 6B2, 7B2, ) - physically overlapping genetically clustered markers are not acceptable. Genetic map should be improved - distal fragment of 3B is inverted - map for all chromosomes from D genome needs thorough revision (information on distortion of segregation should be provided to confirm that distorted markers were removed). Additional association mapping to physical map should be performed to validate QTLs.
  2. no data for yield are presented and TGW is just a component of weight of kernels from spike. Interaction between TGW and yield can be variable and normally all measured components should be presented in single report together with yield data. Transferability of QTLs to different genetic background is not obvious. Therefore practical impact of QTLs is not proved and no candidate genes are proposed - this justify "low significance of content"; To correct this more yield related traits should be included into publication and candidate genes should be proposed.

Loci for grain yield are not presented in manuscript - we do not know how TGW QTLs affect yield – title not fits with contents. Line 185 – no evidence for translocation is provided and generally, for Am better target will be A genome and not D. Data in Tables S2 and S4 not fits - single Table with both genetic and physical for the same set of bin markers should be provided.

Author Response

Dear reviewer,

Thank you very much for valuable suggestions on our manuscript ijms-781755. We have read every points of the valuable corrections carefully, they are reasonable and very helpful to improve the manuscript. We accepted all the corrections and added the information you requested in the manuscript.

Thank you very much for your work on our manuscript!

Sincerely yours,

Dr. Bo Zhang & Huaigang Zhang

 The modification instructions are shown below:

The manuscript provides new data on QTLs related to thousand grain weight (TGW) in wheat. Generally, approach of QTL mapping on biparental DH population, phenotyping in multiple locations and chip-based genotyping are good and promising. However, the main pitfalls are:

Point 1: Low number of plants (~85) reduces precision of genetic mapping and in effect discrepancies between genetic and physical maps are so significant that no candidate genes can be proposed; to minimize these effects construction of genetic map should be revised and generally most of markers not colinear with physical map should be possibly removed (7A2, 1B2, 6B2, 7B2, ) - physically overlapping genetically clustered markers are not acceptable. Genetic map should be improved - distal fragment of 3B is inverted - map for all chromosomes from D genome needs thorough revision (information on distortion of segregation should be provided to confirm that distorted markers were removed). Additional association mapping to physical map should be performed to validate QTLs.

Response 1: Thank you for your suggestion. We redrew all of the genetic maps (Fig. 2) and physical maps (Supplemental Fig.1) after reanalysis the data (markers not co-linear with physical map and physically overlapping genetically clustered markers were deleted). The information about missing rate and Chi-square of all markers were added in Supplemental Table 3. Distal fragment of 3B and all chromosomes from D genome appeared to be normal after redrew the map.

Point 2: No data for yield are presented and TGW is just a component of weight of kernels from spike. Interaction between TGW and yield can be variable and normally all measured components should be presented in single report together with yield data. Transferability of QTLs to different genetic background is not obvious. Therefore, practical impact of QTLs is not proved and no candidate genes are proposed - this justify "low significance of content"; To correct this more yield related traits should be included into publication and candidate genes should be proposed.

Response 2: Thanks for your reasonable suggestion. We have no additional yield-related data, so the title “Mapping quantitative trait loci for grain yield in a double haploid population of common wheat” was changed as “Mapping quantitative trait loci for 1000-grain weight in a double haploid population of common wheat”. And after reanalysis, TraesCS6A01G343600 encodes the E3 ubiquitin-protein ligase same as the TaGW2, it may be considered as a potential candidate gene for qTgw.nwipb-6AL.

Point 3: Loci for grain yield are not presented in manuscript - we do not know how TGW QTLs affect yield – title not fits with contents. Line 185 – no evidence for translocation is provided and generally, for Am better target will be A genome and not D. Data in Tables S2 and S4 not fits - single Table with both genetic and physical for the same set of bin markers should be provided.

Response 3: The title was changed as “Mapping quantitative trait loci for 1000-grain weight in a double haploid population of common wheat” to fit with content. There is a possibility that translocation might have occurred on chromosomes 4D because of the distant hybridization which resulted in the variation that was retained in the DH population. Similar result was obtained in the same DH population which was mapped a stripe rust QTL on 7DS, the resistance is actually originated from T. monococcum 10-1 (AmAm). We added the physical position in the Table S2.

Reviewer 3 Report

In the present study, several QTLs for TGW were detected on a high-density genetic map. The results obtained in the DH population will be valuable because the individuals of the DH population have homozygous genetic background. However, as mentioned by authors in the introduction, there were many reports on TGW in wheat. Are only qTgw.nwipb-4DS and qTgw.nwipb-6AL firstly reported in this paper? Summarize the positions of QTLs previously reported, and if the QTLs detected on the same position of previously reported, indicate the information in Table 3.

L255 mutations? I think this should be loci.

Author Response

Point 1: In the present study, several QTLs for TGW were detected on a high-density genetic map. The results obtained in the DH population will be valuable because the individuals of the DH population have homozygous genetic background. However, as mentioned by authors in the introduction, there were many reports on TGW in wheat. Are only qTgw.nwipb-4DS and qTgw.nwipb-6AL firstly reported in this paper? Summarize the positions of QTLs previously reported, and if the QTLs detected on the same position of previously reported, indicate the information in Table 3.

Response 1: Thanks for your reasonable suggestion. By retrieving, we found that the qTgw.nwipb-1AS, qTgw.nwipb-4BS, and qTgw.nwipb-5AL may be similar to the previously reported. qTgw.nwipb-2DS, qTgw.nwipb-2DL, qTgw.nwipb-4DS, qTgw.nwipb-5DS, qTgw.nwipb-6AL, and qTgw.nwipb-6DS may be firstly reported in this paper. And the information was added in Table 3.

Point 2: L255 mutations? I think this should be loci.

Response 2: Thank you for the suggestion. We changed “mutations” into “loci” in line 258.

Round 2

Reviewer 2 Report

The genetic map has been recalculated and significantly improved. In result some (4) new QLTs were identified and some (3) were lost. New stable QTL on 6A was identified. Physical position of EF447275.1 corresponding to TaGW2-6A should compared with physical position of the QTL qTgw.nwipb-6AL. Comparison of genetic position of TaGW2 with physical position of the QTL is not sufficient. What LOD threshold value was calculated by permutations?

Translocation is good explanation of supressed recombination in region 4D of M321 but some cytogenetic evidence would be profitable to support this hypothesis.

Author Response

Point 1: The genetic map has been recalculated and significantly improved. In result some (4) new QLTs were identified and some (3) were lost. New stable QTL on 6A was identified. Physical position of EF447275.1 corresponding to TaGW2-6A should compared with physical position of the QTL qTgw.nwipb-6AL. Comparison of genetic position of TaGW2 with physical position of the QTL is not sufficient. What LOD threshold value was calculated by permutations?

Response 1: Thanks for your reasonable suggestion. By retrieving the physical location of TaGW2, we found it was located from 237,735,006 bp to 237,759,304 bp on chromosome 6A, which was added in lines 182-183. After 1000 times permutation test, the LOD threshold of BIP module and MET module were 3.22 and 6.23, respectively, the results were added in line 113 and lines 138-139.

Point 2: Translocation is good explanation of suppressed recombination in region 4D of M321 but some cytogenetic evidence would be profitable to support this hypothesis.

Response 2: Thanks for your advice. We will search the cytogenetic evidence for this hypothesis in future.
